# The impact of flipping class intervention on reading comprehension: Different approaches and proficiency levels

Arfan Fahmi[1,2], Nur Mukminatien[1], Daniel Ginting[3]*, Shirly Rizki Kusumaningrum[1]

1 Universitas Negeri Malang, Malang, Indonesia, 2 Institut Teknologi Sepuluh Nopember, Surabaya, Indonesia, 3 Universitas Ma Chung, Malang, Indonesia

* daniel.ginting@machung.ac.id

**Data Availability Statement:** All relevant data are within the manuscript and its Supporting Information files. This is the link for you to get the data: 10.6084/m9.figshare.25827178.

## Abstract

Flipped classes can improve English language teaching and learning, especially reading outcomes, by enhancing student engagement and motivation through interactive and captivating educational materials. The goal of this study is to examine the effects of different schema activations in different English flipped class formats in pre-reading activities on the reading comprehension of students with different reading proficiency levels. A quasi-experimental research design was employed involving 30 first-year students from the first university and 28 first-year students from the second university in Surabaya, Indonesia, who were studying English as a general course at an intermediate level. The study used informed consent and two different formats, A and B, where pre-reading tasks were completed in class or asynchronously online, respectively. The study found that using video-based pre-reading activities in flipped class can improve comprehension and schema acquisition. Flipped classes can provide personalized learning experiences, but its effectiveness varies depending on the strategies and delivery methods used. While most students benefit from flipped classes, those who struggle with self-discipline and time management may find it challenging to adapt to the online component and may experience lower performance as a result.

## Introduction

Researchers are increasingly interested in the flipped classroom model as an innovative instruction approach to improve English language teaching and learning [1–3]. While conventional teaching relies on in-class lectures, the flipped model emphasizes flexibility, autonomy, and collaborative activities during face-to-face sessions. The flipped classroom involves students watching pre-recorded video lectures or reading materials at home, and then completing interactive activities or assignments during class time. Kazakoff et al. [2] found that flipped classes could improve reading outcomes, particularly among children who struggle with reading. The students who have problems with reading skills need more time to comprehend and practice. They can access learning materials and activities at their own pace and on their own timetable to comprehend and process the content [4–7]. Consequently, students work through the material at a comfortable speed with greater control over their education.

**Funding:** The author(s) received no specific funding for this work.

**Competing interests:** The authors of this manuscript have thoroughly reviewed the journal's policy on competing interests, and we unequivocally declare that there are no competing interests associated with this work. We affirm that neither financial nor non-financial interests exist that could be perceived to influence or bias the outcomes or interpretations presented in this research. Our commitment to transparency and integrity in research is paramount, and we affirm that there are no relationships or activities that could compromise the impartiality or objectivity of this study. This statement is in line with the journal's guidelines, and we assure the readers and the scientific community of the accuracy and completeness of this declaration.

In comparison to traditional teaching methods, flipped classes demonstrate a unique advantage in boosting student engagement and motivation. While both approaches utilize educational materials, the captivating and interactive nature of online resources, such as games, simulations, virtual labs, and films [8, 9], is a distinctive feature of flipped learning. Unlike traditional methods, flipped classes offer students immediate feedback on their reading progress, allowing them to identify specific areas for development and measure their growth over time. Samarakou et al. [10] concluded that mobile application-based language learning (MALL), a component of the flipped class approach, outperforms more conventional methods in enhancing reading comprehension in English as a foreign or second language (EFL/ESL). This suggests that the innovative use of technology within the flipped model provides a unique advantage over traditional approaches. Additionally, the potential of flipped classes as an innovative approach to improving English language teaching and learning has been recognized by Lu et al. and Thomas and Plaspohl [1, 3] also noted the potential of flipped classes as an innovative approach to improving English language teaching and learning. Contrasting with conventional methods using similar materials, the flipped approach demonstrates its efficacy in leveraging interactive and technology-driven resources, providing students with a more engaging and feedback-rich learning experience.

Nevertheless, other studies show failures of flipped class programs if they are badly designed and implemented. For example, Rong and Choi [11] found that teachers might be unable to integrate technology and online instruction into their curriculum without proper training and support. Similarly, if teachers lack the necessary training and resources, they may struggle to effectively incorporate technology and online instruction into their teaching practices [12–14]. The online instruction might be poorly designed, for example, difficult to navigate, have unclear instructions, lack interactive elements, or not be age-appropriate. In short, poor design, poor implementation and poor alignment with student needs and interests leads to poor outcomes for students, such as frustration, disengagement and a lack of progress [11, 15, 16].

Flipped class programs also need reliable technology and internet access, which can be hard for some schools and students to get. Frank et al. [17] mention that flipped class depend on students and teachers having reliable and adequate access to technology and the internet. Without it, students and teachers may not be able to take full advantage of the opportunities provided by flipped classes [16, 18].

More research is needed to find out if the flipped classroom is a good way to help students understand what they are reading over time. The goal of this study is to fill in this gap by looking at how different types of flipped class and pre-reading activities with different levels of reading skill affect reading comprehension. Specifically, the study seeks to answer the research question: Does a flipped class format improve the learning outcomes in reading comprehension for Indonesian university students for intermediate English EFL, as measured by TOEFL test questions? The researchers compared the results to other studies of flipped classes, and proposed hypotheses that might explain the inconsistency of outcomes.

## Literature review

### Schema activation

A schema is a mental framework that represents a person's knowledge or understanding of a particular concept, category, or event. Meanwhile, schema activation refers to the process by which a mental framework or schema is brought to mind and used to interpret and organize incoming information [19]. When a schema is activated, it can influence how a person perceives, processes, and remembers information that is related to that schema. Studies suggest that flipped class can improve schema activation in reading by providing multimedia resources

and interactive activities that engage students in the learning process. According to Kuo et al. [20], flipped class can enhance schema activation in reading through interactive activities and multimedia resources that engage students. Similarly, Xu et al. [21] found that virtual learning environments in flipped class can encourage schema activation through collaborative and interactive learning. By integrating multimedia resources, interactive activities, and collaborative virtual environments, flipped classrooms create an enriched learning environment that actively supports the activation of cognitive frameworks essential for robust reading comprehension.

However, not all studies support the use of technology in promoting schema activation in reading. Some research suggests that it has a detrimental impact. For instance, Mangen et al. [22] found that students who read texts on computer screens activated fewer schemas than those who read the same texts on paper. Having a summary beforehand was the cause, Chen and Lo [23] investigated the relationship between reading skills, computer use skills, website features, and online reading performance. They found that reading and computer use skills were critical for successful online reading, and specific website features, such as clear layout and helpful navigation, could promote schema activation and comprehension. Cohen and Kenny [24] explored the effect of digital media on reading processes, including schema activation, and argued that, while digital media could provide new opportunities for engagement and learning, it could negatively affect attention, memory, and comprehension. Thus, we conclude that the impact of technology on schema activation is intricate. It implies that acknowledging the intricacy and varied influences on schema activation due to technology use underscores the importance of tailored educational approaches, digital literacy, thoughtful learning environment design, balancing advantages and disadvantages, and ongoing research to inform effective educational practices. Cho and Ma [25] investigated the impact of e-text features, such as highlighting and note-taking, on reading comprehension and schema activation. They found that certain features, such as interactive glossaries and hyperlinks to related content, could promote schema activation and comprehension. Moreover, Seong et al. [26] examined the effects of text difficulty, prior knowledge, and reading mode (paper vs. screen) on cognitive load and comprehension. The findings suggest that reading on a screen can increase cognitive load, but this effect could be mitigated by using interactive features that promote schema activation and comprehension. Liu et al. [27] investigated the impact of digital annotation (highlighting and note-taking) on second language (L2) reading comprehension. The study found that digital annotation could improve L2 reading comprehension by helping learners activate their prior knowledge and focus on critical information in the text. On the other hand, Skains [28] investigated the effects of hyperlinks on text comprehension and cognitive load in English as a foreign language (EFL) reading. The study found that hyperlinks could promote comprehension and schema activation but could increase cognitive load if they were too distracting or numerous. Moreover, Maaranen and Kynäslahti [29] explored the potential of multimodal digital reading pedagogies (which incorporate a range of media, including text, images, audio, and video) for developing digital literacies. They argued that such pedagogies could support schema activation and comprehension by presenting information in multiple modalities.

The impact of digital media on reading processes is still being explored, and while it can offer new opportunities for engagement and learning, it may also have negative effects. Consequently, it is crucial to consider the design of digital media and its features to support schema activation and comprehension while minimizing potential negative effects such as increased cognitive load. The studies reviewed in this synthesis highlight the importance of schema activation in comprehension and how it can be impacted by flipped class and digital media. They provide valuable insights into the potential benefits and drawbacks of integrating technology

into education and emphasize the need to carefully consider the design and features of digital media to support schema activation and comprehension.

### Flipped class

Flipped classrooms have been shown to increase student engagement and provide more opportunities for personalized learning [30]. In each of the following studies, researchers assigned students to either a traditional or flipped classroom group and found that the students in the flipped classroom group had significantly higher reading comprehension scores compared to the traditional group:

1. Brown and Czerniewicz [31] had a group of 30 higher education students. This study involving higher education students provides an opportunity for in-depth analysis due to the maturity of participants

2. Lo and Hew [32] studied a group of 120 fourth-grade students. This research with fourth-grade students is relevant to K-12 settings and offers insights into long-term academic impacts, yet its applicability to higher education and age-related considerations pose limitations

3. Kuo et al. [20] had group of 63 non-native English speakers. They focused on non-native English speakers brings diversity to the participant pool and relevance to language learning contexts.

   Common considerations across all studies include relatively small sample sizes, varied durations of interventions, and different measures of reading comprehension, impacting the comparability of results. Nevertheless, these studies indicate that the flipped classroom model is a promising approach for educators seeking to enhance their students' reading skills. It has been shown to increase student engagement and provide more opportunities for personalized learning

## Materials and methods

### Research design

This study involved 30 first-year students from the first university and 28 freshmen from the second university in Surabaya, Indonesia. The research design was quasi-experimental because it was not feasible to randomly assign participants to groups. Although studying at different universities, these students had schedules officially set by university administrators, and similar time constraints. We acknowledge its limitations, particularly the potential for selection bias due to the non-random assignment, which may impact the generalizability of the findings. Both groups also had similar characteristics; they came from non-English departments, studied English as a general course, and had the same intermediate level of English proficiency. The similarities among participants were carefully assessed and ensured through several key steps in the research methodology. Firstly, during the participant selection process, specific criteria were established to include only first-year students from both universities who were enrolled in non-English departments, attended English as a general course, and demonstrated an intermediate level of English proficiency. This ensured that the selected participants shared common academic contexts. Furthermore, university administrators officially set the schedules for all students, regardless of the university they attended, contributing to similar time constraints. This administrative control helped standardize external factors that could influence the study outcomes.

This research received ethical approval from the Institutional Review Board (IRB) of the Fakultas Desain Kreatif dan Bisnis Digital Departemen Studi Pembangunan, Institut Teknologi Sepuluh Nopember, in December 2022. The IRB reviewed the application in accordance with regulatory standards and determined that the study is exempt from further IRB review. The informed consent protocol was implemented from January 15 to January 20, 2023. The study strictly adhered to ethical guidelines, including principles of informed consent, confidentiality, and voluntary participation. The voluntary nature of the research was emphasized, and participants were assured of their freedom to withdraw from the study at any point. In the meeting, the researcher explained the content of the consent form, ensuring participants' full comprehension of the research's nature and implications. Privacy was safeguarded through the use of pseudonyms. Participants were given the opportunity to ask questions before providing their written consent. They signed the consent forms, indicating their voluntary agreement to participate in the study. Written consent was obtained from all participants after presenting them with a detailed consent form.

In this study, the researchers used two different formats (A and B) for teaching reading skills to participants. In format A, the participants completed pre-reading and while-reading activities in class, while the post-reading exercises were done online outside of class. There were three sets of readings with pre-reading exercises that included activities such as watching videos and completing matching and missing word exercises. During class, the while-reading activities included responding to comprehension questions and correcting factual errors. After class, the participants completed online post-reading exercises, such as writing descriptive paragraphs. The post-test was given at the end of the sixth week. In format B, the researchers flipped the class, meaning the pre-reading exercises were completed online before class, while the while-reading activities were done face-to-face in class. The post-reading tasks were also completed online outside of class. The pre-reading exercises included watching a video, responding to questions, and completing matching and missing word exercises. The while-reading activities were similar to those in format A, and participants were encouraged to use additional online resources to improve their understanding.

Finally, online post-reading exercises are provided, including writing comparisons and descriptive paragraphs. In format B, the participants engaged in online pre-reading and in-class activities with text 1 during week 1. During the second week, students were involved in reading activities with text 2 in class and post-reading activities online. The activities of weeks 3 and 5 mirrored those of week 1, while those of weeks 4 and 6 repeated those of week 2. The reading activities in Format A and Format B are simplified on in Table 1. Like the previous group, we gave the participants the post-test in the 6th week.

**Table 1. The reading activities in format A and format B.**

| Format A | Format B |
| --- | --- |
| Three packs of intensive reading instructions with two reading texts. | Three packs of intensive reading instructions with two reading texts |
| Pre-reading activities<br>Watching short videos and answering questions about them, completing missing words, and matching | Pre-reading activities<br>Asynchronous activities: watching a short video and answering quizzes online |
| While-reading activities<br>Reading activities in class | While-reading activities<br>Reading activities in class |
| Post-reading activities<br>Post-reading online activities: writing comparisons and descriptive paragraphs | Post-reading activities<br>Post-reading online activities: writing comparisons and descriptive paragraphs<br>descriptive paragraphs are given online |

The chosen activities in formats A and B aim to optimize reading comprehension. In format A, class-based pre-reading tasks, such as video-watching and exercises, set the foundation. While-reading activities, like comprehension questions, reinforce understanding in class, followed by online post-reading exercises. In format B, a flipped class model shifts pre-reading online, fostering individual preparation. Face-to-face while-reading activities encourage engagement, with post-reading tasks completed online. This balanced approach, combining in-person and online elements, seeks to activate prior knowledge, promote real-time comprehension, and provide opportunities for practical application, aiming for a lasting improvement in reading comprehension skills.

## Sample and data collection

Before collecting data, we conducted a pilot test to ensure that the research procedure and assessment tool were accurate in gauging participants' reading competency and proficiency levels. We selected a small sample of participants with similar characteristics to those who would be included in the actual study. We asked them to take the tests in language laboratories. These participants were given both Test A and Test B of the TOEFL ITP® Assessment Series to gauge their reading competency and proficiency levels. From these findings, we made some necessary adjustments to enhance the study's overall design, ensuring a more robust and efficient data collection process in the actual study. For example, we observed that the implementation of the pilot test was quite satisfactory: all test takers could complete the test based on the test protocol, and we could also get the data to analyse. However, test schedules had to be adjusted due to computer availability in the language laboratories. Over all, we observed that the implementation of the pilot test was quite satisfactory: all test takers could complete the test based on the test protocol, and we could also get the data to analyse. Test schedules had to be adjusted due to computer availability in the language laboratories.

The reading comprehension element of the TOEFL ITP® Assessment Series, which consists of 2 sets of tests, is used to gauge participants' reading competency at the start of implementation and reading comprehension after each flipped class intervention (Test A and Test B). The choice of the TOEFL ITP® Assessment Series was deliberate and aligns closely with the specific objectives of the study. The reading comprehension tests (Test A and Test B) were selected because they assess skills synonymous with the learning outcomes targeted in the flipped class interventions. These skills include the ability to identify main ideas, extract specific information, and comprehend implicit details—skills emphasized within the flipped classroom model. The mapping of the TOEFL® ITP tests onto the Common European Framework of Reference provided a structured framework for categorizing reading proficiency levels. This alignment ensures that the assessment tool effectively captures the nuanced improvements in reading comprehension targeted by the flipped class interventions

Moreover, the TOEFL ITP® Assessment Series examinations share traits with the skills taught in the flipped class, such as the need to recognize the primary idea, uncover the specifics of a statement, and comprehend implicit information. Educational Testing Service (ETS) also gave the writers a grant to print and electronically replicate the material on Request#38286. Based on the TOEFL® ITP Tests Mapping onto the Common European Framework of Reference, the divide was made according to the reading proficiency level. These are the three levels' descriptions:

˚ HIGH = ≥56 (ITP TOEFL Reading Comprehension section)

˚ MID = 48≤55 (ITP TOEFL Reading Comprehension section)

˚ LOW = 31≤47 (ITP TOEFL Reading Comprehension section)

Source: Mapping the TOEFL® ITP Tests onto the Common European Framework of Reference (ETS, 2012).

We selected two participants for an interview, and their perspectives were intended to strengthen our findings in the field. Henry (pseudonym) belonged to the class that we taught using method A, and Julia (pseudonym) is a participant who was taught using method B and invited to participate in a private location with necessary equipment. The selection criteria were meticulously crafted to ensure comprehensive representation from both instructional methods (A and B). This involved considering factors such as participant performance, engagement levels, and demographic characteristics to guarantee a diverse and well-rounded sample. By intentionally selecting participants from each method, we aimed to capture a holistic perspective on the effectiveness of both instructional approaches, enriching the depth and breadth of insights gathered during the interviews. Consent was obtained, and the interviewer asked open-ended questions to encourage free sharing of thoughts and experiences. Confidentiality was maintained. The interview was transcribed and analysed to develop conclusions and recommendations based on the research questions.

## Analyzing of data

Because there were less than 30 participants in each flipped class format (Format A vs. Format B), all scores were examined using non-parametric statistical methods. The effectiveness of each format both before and after the treatment was examined using an independent Sample t-Test. To compare the results of the low, mid, and high proficiency subjects in Format A and Format B, the authors used a Mann-Whitney test.

The utilization of non-parametric statistical methods in our study, necessitated by the limited sample size in each flipped class format (Format A vs. Format B), introduces inherent constraints. Small sample sizes may compromise the generalizability of findings, raising concerns about extrapolating the results to broader educational contexts.

Moreover, applying an independent Sample t-Test to evaluate the effectiveness of each format before and after treatment introduces limitations due to assumptions of normal distribution, despite our overall reliance on non-parametric tests. This inconsistency may impact the robustness of our findings, particularly in capturing nuanced changes within each format over time. Last, The Mann-Whitney test's use to compare results among proficiency groups in Format A and Format B addresses non-normality concerns but is constrained by the small sample size, impacting result reliability. The variability within these proficiency groups may not be fully accounted for, influencing the outcomes of the Mann-Whitney test. While non-parametric methods were chosen given the sample size contraints, the study acknowledges their limitations. Caution is advised in generalizing the findings beyond the specific sample and contexts studied.

## Statistical validations

In this study, we gave a pre-test and post-test to the participants. These tests allowed us to track changes over time, which could be used to determine the impact of the treatment. The experiment's first step was administering a reading proficiency pre-test on the 30 first-year students from the first university and the 28 from the second university who were enrolled in the non-English Language Education Department. We employed this test to gain a baseline measurement against which the post-test results can be compared. The outcomes are shown in Table 2.

Table 2 shows that the mid-level (N = 29 or 50%) predominated the participants, followed by the low and high levels. Additionally, as indicated in Table 3, an Independent Sample t-Test

**Table 2. The reading proficiency level of the participants.**

| No | Level | 1st University | 2nd University | Total |
|---|---|---|---|---|
| 1 | Low | 8 | 12 | 20 |
| 2 | Mid | 16 | 13 | 29 |
| 3 | High | 6 | 3 | 9 |
| Total | | 30 | 28 | 58 |

was performed to determine whether participants from the two universities had various proficiency levels before the flipped class treatment.

The significance level (P-Value) reported in Table 3 is higher than .05, indicating no statistically significant difference between the participants' reading abilities in the two universities before the treatment. It means that the difference observed between the groups or measures is likely due to random assignment, not a real difference. Both groups of participants had comparable reading abilities before the intervention, which was important to commence the process. After running the reading program for six weeks, we conducted a post-test. The effects of flipped class utilizing Formats A and B were tested on the individuals again after the treatment. Table 4 compares the participants' results from the two groups' reading proficiency tests.

Table 4 demonstrates that the P-Value achieved was .000, which is less than .05. It indicates that the learning outcomes for individuals in Format A and Format B differed. Both formats' results scores were compared. Participants who received Format A had a median score of 51.50, whereas those who received Format B had a median score of 54.00. The Mann-Whitney test result was .00 with a 95.03% confidence level. The flipped class had a very different impact on Formats A and B, with Format B outperforming Format A. Last, it was crucial to evaluate the outcomes of contrasting the scores attained by the participants with low, mid, and high competence levels in Format A and Format B, as shown in Table 5. Overall, the findings suggest that flipped class had a significant impact on learning outcomes, with Format B being more effective than Format A.

The findings from Table 5 suggest that the effectiveness of Format A and Format B differed across different levels of reading competency. Participants with limited reading competency who received Format B outperformed those who received Format A, as indicated by the higher median score of 53.00 for Format B compared to 49.00 for Format A. The same pattern was observed for participants with mid-reading competency, with a higher mean score of 55.00 for Format B compared to 52.00 for Format A. For participants with good reading proficiency, the median score for Format B was higher than that of Format A (58.00 compared to 56.00). However, the difference between the two formats was not statistically significant at the 95.77%

**Table 3. The result of the reading proficiency test before the treatment.**

| No | Participant | N | Min | Max | Median | P-Value | Achieved Confidence |
|---|---|---|---|---|---|---|---|
| 1 | 1st University | 30 | 44 | 58 | 54.00 | .06 | 95.05% |
| 2 | 2nd University | 28 | 43 | 56 | 52.50 | | |

**Table 4. The result of the comparison of scores obtained by the participants in format A and format B.**

| No | Subjects | N | Median | P-Value | Achieved Confidence |
|---|---|---|---|---|---|
| 1 | Format A | 58 | 51.50 | .000 | 95.03% |
| 2 | Format B | 58 | 54.00 | | |

**Table 5. The result of the comparison of scores obtained by the subjects with low, mid, and high proficiency levels in format A and format B.**

| Proficiency | Format | N | Median | P-Value | Achieved Confidence |
|---|---|---|---|---|---|
| Low | A | 20 | 49.00 | .000 | 95.01% |
| | B | 20 | 53.00 | | |
| Mid | A | 29 | 52.00 | .000 | 95.17% |
| | B | 29 | 55.00 | | |
| High | A | 9 | 56.00 | .013 | 95.77% |
| | B | 9 | 58.00 | | |

confidence level, with a Mann-Whitney test result of .013. Overall, the findings suggest that Format B was more effective in improving reading comprehension outcomes than Format A, particularly for participants with limited or mid-level reading competency.

Table 5 compares scores across low, mid, and high proficiency levels in Format A and Format B. For low and mid proficiency, Format B demonstrates significantly higher median scores (53.00 and 55.00, respectively) than Format A (49.00 and 52.00), supported by P-values of .000 at confidence levels above 95%. However, for high proficiency, while Format B (58.00) has a higher median than Format A (56.00), the difference is not statistically significant (P-value = .013) at the 95.77% confidence level. The findings suggest that flipping class interventions can have different effects on learners' reading comprehension outcomes depending on the format of the intervention and the learners' proficiency level. In this study, Format B was found to be more effective than Format A in improving learners' reading comprehension scores overall and among learners with limited, mid, and high reading competency levels.

These results have implications for educators and instructional designers who are considering implementing flipping class interventions to improve learners' reading comprehension. Careful consideration should be given to the format of the intervention and the learners' proficiency levels when designing and implementing flipping class interventions. It may be necessary to tailor the interventions to the learners' proficiency levels to achieve the desired outcomes. Based on the research findings, it appears that different formats of flipped class interventions may have varying impacts on learners' reading comprehension, and these impacts can also differ based on the learners' proficiency levels. For instance, if learners have limited reading skills, a flipping class intervention may need to focus more on building basic reading skills and providing additional support to help learners comprehend the material. Similarly, if learners have more advanced reading skills, a flipped class intervention may need to incorporate more challenging material and activities to maintain engagement and further enhance their reading comprehension

## Results and discussions

### Video-based pre-reading improves comprehension

We found that the students involved in format B (both pre-and post-reading online activities) performed better than those in format A (on-site pre-teaching and online post-reading activities). In Format B, the pre-teaching activities started by sending videos to the students before face-to-face sessions in class.

Julia, who used method B, said the following in an interview.

"*I just found out that using videos before class can help me learn better. I'm really excited about this*! *I can watch the videos at my own speed and feel like it will help me do better in my*

*classes. I'm going to start incorporating video learning into my study routine to see if it helps me improve my grades."*

Henry who belonged to group A commented as the follows:

"*I have some concerns about offline pre-reading activities as I find them quite monotonous and uninspiring. Personally, I feel like I must complete a task before I can start learning something new. I believe it would be more beneficial for me to engage with more creative and interactive ways of preparing for class. Perhaps teachers can consider finding more innovative and stimulating approaches to support our learning process*".

We presumed that videos presented in format B are a powerful tool for activating students' schema to learn new concepts before class, leading to better reading comprehension skills [33, 34]. Videos summarize key vocabulary, preview the material's structure, and provide real-world examples, making the material more engaging and relevant to students [35]. Using videos as part of a flipped class approach, that is, format B, can activate students' prior knowledge and experiences about a topic.

Videos offer a visually engaging way for students to interact with the subject matter. The goal is to cultivate increased participation, interest in related activities, and a stronger connection between students and the material, ultimately making them more receptive to new information during subsequent in-class sessions. Moreover, the videos summarized key vocabulary, previewed the material's structure, and provided real-world examples, making the material more engaging and relevant to students. This increased engagement and relevance can lead to better retention and recall of the material.

Moreover, in the flipped class, students absorb information at a rate that suits their individual understanding, preventing an overwhelming influx of new content. They can allocate cognitive resources more efficiently, fostering a deeper understanding of the subject matter. Thus, this reduction in cognitive load enhances the learning experience, promoting a more seamless and effective acquisition of knowledge [36, 37]. Our findings align with existing research supporting the idea that pre-reading activities, like video-based approaches, can activate relevant schemas, contributing to enhanced comprehension, retention, and recall of information [38–40].

## Flipped class improves personalized outcomes

We found that flipped learning offers "personalized outcomes." That is, the approach can be tailored to meet the individual needs of students, allowing them to learn at their own pace and in their preferred style. By using a blend of different learning methods, flipped class can offer students a more customized and engaging learning experience that can lead to better learning outcomes. A more personalized and engaging way through flipped class offers students the flexibility to learn and access a variety of learning materials and resources, such as videos, interactive simulations, and online assessments. By using these resources, students can engage with the material in a way that is most effective and comfortable for them. This personalized learning helps students stay motivated and engaged, leading to better learning outcomes.

Julia joining in the flipped class has shared her testimony.

"*I'm really into this method! It's perfect for me because I can learn in the way I like best. Plus, there are so many cool things to use like videos, interactive simulations, and online assessments that help me really get into the material. My teachers can even keep tabs on how I'm doing and adjust their teaching style to fit my needs. It's awesome to get this kind of*

*personalized attention and feedback—it really keeps me motivated and engaged, and helps me learn better in the end!" Top of Form*

Furthermore, our finding that flipped class approaches, such as format B, outperform traditional methods (format A) is consistent with other researchers' findings [31, 32]. For example, a study by Kuo et al. [20] found that using a flipped classroom approach improved English reading comprehension for non-native speakers. Similarly, a study by Lo and Hew [32] showed that a flipped classroom approach improved reading comprehension scores for primary school students. These studies support the idea that flipped class approaches can provide a more personalized learning experience and improve student learning outcomes.

Thus, our findings suggest that incorporating pre-teaching videos in format B can be an effective strategy for promoting students' reading comprehension skills. This aligns with other studies that have found that pre-reading activities can activate relevant schemas and support comprehension, retention, and recall of information. Additionally, our study's finding that a flipped class approach (format B) outperformed traditional methods is consistent with other research in the field. These findings provide important insights for educators seeking to improve students' reading comprehension skills and enhance their overall learning outcomes

## Delivery and strategies affect pre-reading efficacy

Our finding has shown that activating relevant schemas through the online platform is efficient in supporting comprehension and schema acquisition which aligns with schema activation theory: activating relevant schemas before reading can help readers make connections between new information and their existing knowledge, improving comprehension and retention of information.

Julia gives her testimony about the video shared in the online pre-teaching.

*"Honestly, the videos that teachers share for online pre-reading are pretty cool. They explain new stuff and topics. But sometimes, I get pretty bored with things I already know. Like, the other day my teacher made us watch a video on how to open a bank account. I already knew all that stuff, so it didn't really help me out".*

In the case above, the student is already familiar with the material being presented in the pre-reading activities, either because they have already studied it in the past or because they have a strong background in the topic. As a result, the pre-reading activities may not be challenging or engaging enough for the student, and they may find them repetitive and unhelpful for their learning. Thus, if he has already taken a course in finance and has a strong understanding of financial concepts, they may find pre-reading activities on the basics of financial management to be too simplistic and not worth their time. Similarly, if a student has a strong interest in a particular topic and has already done extensive reading on the subject, they may find pre-reading activities on the same topic to be redundant and not useful for their learning. In such cases, the student may benefit from more advanced or challenging pre-reading activities that build upon their existing knowledge and skills. Alternatively, the student may choose to skip the pre-reading activities and focus on other aspects of the course that are more relevant or challenging for their learning

Nevertheless, some other studies found that pre-reading activities using videos did not bring about positive advantages in schema acquisition compared to traditional reading instruction. [41], for example, investigated the effectiveness of pre-reading activities using videos in a flipped class environment on EFL learners' reading comprehension and found that

pre-reading using videos did not bring about positive advantages in schema acquisition compared to traditional reading instruction. The study involved two groups of students: one group was assigned to watch a series of videos as pre-reading materials, while the other group was assigned to read the same materials without watching the videos. The researchers found that there was no significant difference between the two groups in terms of their reading comprehension scores. This suggests that pre-reading using videos (flipped class) does not bring about positive advantages in schema acquisition. Another study by Lai and Chen [42] also reported that pre-reading using videos did not have a significant effect on learners' reading comprehension in a Taiwanese context. They examined the effect of using videos as pre-reading materials on reading comprehension in a college EFL class in Taiwan. The study involved two groups of students: one group was assigned to watch videos before reading the text, while the other group read the text without watching the videos. The researchers found that there was no significant difference between the two groups in terms of their reading comprehension scores. This study also suggests that pre-reading using videos may not be effective in promoting schema acquisition.

We conclude that these opposing views suggest that the effectiveness of pre-reading activities may depend on the mode of delivery and the specific instructional strategies used. It is possible that the videos used in these studies did not effectively activate the relevant schemas for the learners, leading to no significant differences in comprehension scores. Overall, the findings of these studies suggest that the mode of delivery and the specific instructional strategies used for pre-reading activities can influence the effectiveness of schema activation in supporting comprehension and retention of information.

## Flipped class effectiveness varies by strategy

In our study, we found several cases where some other participants still struggle with self-discipline because the flipped class offers a lot of benefits to most participants. These students still struggle to adapt to the online component of the program, particularly if they lack self-discipline and time management skills. Some participants in our study struggled to adapt to the online component because they lacked self-discipline and time management skills. They miss deadlines, or struggle to focus and stay engaged in the online environment. As a result, their performance may suffer, even if they excel in face-to-face instruction. This is because traditional classroom instruction provides more structure and support, which can help these students stay on track and engaged. Julia has shared her experience with us.

Julia gives her testimony about her self-disciplines.

*So, my English teacher set up this new way of learning where I only go to school twice a week and do the rest of the work online. But I'm having trouble keeping up with the deadlines for the assignments. I get distracted by social media and other stuff online, and sometimes I just don't feel like doing the work. Like, I'll think to myself, "I can do this later," and then I end up playing games or doing something else instead.*

Similarly, van Alten et al. [43] also found that, while flipped classes had some positive effects on student learning outcomes, they did not lead to significant improvements in self-regulated learning or student motivation. A meta-analysis by Låg and Sæle [44] found that flipped classes had a small positive effect on student achievement, but that the effect was not statistically significant. A study by Fisher et al. [45] found that while flipped classrooms were associated with higher student satisfaction and engagement, they did not lead to significant improvements in student learning outcomes. A study by Jdaitawi [46] found that

while students in a flipped classroom reported higher levels of engagement and motivation, there were no significant differences in their learning outcomes compared to students in a traditional classroom.

These studies point to the conclusion that the effectiveness of flipped class approaches may depend on various factors, such as the specific instructional strategies used, the mode of delivery, and the student characteristics. Regarding student characteristics, we have found that certain students may need more structure and guidance to stay on track with their coursework. They may procrastinate and wait until the last minute to complete their online coursework. Moreover, they also struggle with distractions, such as social media or other online activities. They may also have difficulty managing their time effectively, particularly if they are also juggling other commitments, such as work or family responsibilities.

## Conclusion

The study concluded that students taught using format B, which included both pre-reading and post-reading online activities, performed better in reading skills compared to students taught using format A, which involved on-site pre-teaching and online post-reading activities. The study found that videos presented during pre-teaching in format B served as a powerful tool for activating students' schema, making the material more interesting and relevant to students and motivating them to learn. The personalized learning experience in format B also allowed students to learn at their own pace, developing self-regulated learning skills. These findings are consistent with other studies that emphasize the importance of schema acquisition for reading comprehension and the impact of self-control and self-regulated learning on learning results. However, some studies do not support the effectiveness of schema activation in pre-reading activities using videos, suggesting that the relationship between video-based pre-reading activities and schema activation may not be straightforward. Future research directions could explore the long-term effects of flipped classroom interventions on reading comprehension, assess applicability across different age groups, and delve into the influence of learner motivation and self-regulation within these interventions.

Based on this study, we propose some implications for educators and policymakers. First, educators in similar contexts can apply the findings by incorporating effective strategies from Format B into their instructional approaches. Key strategies include integrating video-based pre-reading activities to activate students' schema and providing a personalized learning experience that allows students to progress at their own pace, fostering self-regulated learning skills. To replicate these strategies in different educational settings, educators can introduce engaging videos aligned with the curriculum, explore platforms supporting personalized learning, offer teacher training for effective online guidance, adapt strategies to contextual needs, and implement a continuous evaluation process for refinement. This approach ensures a tailored and effective implementation of successful strategies for enhancing reading skills in diverse educational environments. Second, it suggests that more research is needed to better understand the conditions under which online learning is most effective, as well as the factors that may contribute to its success or failure. This can help educators and policymakers make more informed decisions about when and how to use online learning, as well as identify areas for improvement.

Overall, success of online learning shown in some studies but not in others underscores the need for a nuanced and evidence-based approach to education, in which different approaches and strategies are carefully evaluated and tailored to the specific needs of students and contexts

## Limitations

The study involved 30 first-year students from the first university and 28 freshmen from the second university in Surabaya, Indonesia, which may not be representative of the entire population of students. The sample size is small, which may affect the generalizability of the results to other populations. Additionally, the study did not involve random assignment of participants to the experimental and control groups, which may affect the internal validity of the study and limit the ability to establish a cause-and-effect relationship between the flipped class intervention and reading comprehension. The study was also carried out over a six-week period, which may not be sufficient to capture the full impact of the flipped class intervention on reading comprehension. Furthermore, the study only focused on reading comprehension, and the results may not be applicable to other areas of language learning. Finally, the study only involved first-year students from non-English departments who had intermediate levels of English proficiency, and the results may not be applicable to other levels of language proficiency or to students from different departments.

## Author Contributions

**Conceptualization:** Nur Mukminatien, Daniel Ginting.

**Data curation:** Daniel Ginting, Shirly Rizki Kusumaningrum.

**Formal analysis:** Nur Mukminatien, Daniel Ginting, Shirly Rizki Kusumaningrum.

**Methodology:** Arfan Fahmi, Daniel Ginting.

**Validation:** Arfan Fahmi.

**Writing – review & editing:** Arfan Fahmi, Nur Mukminatien.

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
