## [Decision Letter · Decision Letter 0]

20 Feb 2024

PONE-D-23-42134The Impact of Flipping Class Intervention on Reading Comprehension: Different Approaches and Proficiency LevelsPLOS ONE

Dear Dr. Ginting,

Thank you for submitting your manuscript to PLOS ONE. After careful consideration, we feel that it has merit but does not fully meet PLOS ONE’s publication criteria as it currently stands. Therefore, we invite you to submit a revised version of the manuscript that addresses the points raised during the review process.

We look forward to receiving your revised manuscript.

Kind regards,

Daner Sun

Academic Editor

PLOS ONE

Journal Requirements:

Reviewers' comments:

Reviewer's Responses to Questions

**Comments to the Author**

1. Is the manuscript technically sound, and do the data support the conclusions?

Reviewer #1: Yes

Reviewer #2: Partly

2. Has the statistical analysis been performed appropriately and rigorously? 

Reviewer #1: Yes

Reviewer #2: Yes

3. Have the authors made all data underlying the findings in their manuscript fully available?

Reviewer #1: Yes

Reviewer #2: Yes

4. Is the manuscript presented in an intelligible fashion and written in standard English?

Reviewer #1: Yes

Reviewer #2: Yes

5. Review Comments to the Author

Reviewer #1: PONE-D-23-42134

Overall comments:

This study "The Impact of Flipping Class Intervention on Reading Comprehension: Different Approaches and Proficiency Levels" examines an important educational strategy. However, the methodology presents challenges in participant selection and assignment, which could influence the study's validity due to potential biases or lack of representativeness. The generalizability of findings to a wider educational context is not clearly established.

Another big question is that the theoretical framework needs more comprehensive integration with the empirical work, and the analysis does not fully address potential confounding variables. There might be a need for more rigorous statistical testing to strengthen the evidence for the claimed effects in the discussion part. Also, it might be important to also discuss the flipped classroom model's application across diverse educational settings.

Introduction:

1.Row 1, Page 1: While the introduction establishes the relevance of flipped classrooms in language learning, it lacks a direct comparison with traditional methods. Suggestion: Briefly contrast flipped classrooms with conventional teaching approaches to highlight specific benefits or challenges.

2.Row 6, Page 1: The claim that flipped classes improve reading outcomes in struggling readers needs further substantiation. Suggestion: Clarify why this model is particularly effective for these students.

3.Row 12, Page 1: The discussion on engaging materials in flipped classes is good but lacks a comparative aspect. Suggestion: Contrast with traditional methods using similar materials to identify unique benefits of the flipped approach.

4.Row 18, Page 1: The introduction points out potential failures in flipped classroom implementations but needs more specificity. Suggestion: Elaborate on what constitutes effective vs. ineffective design in flipped classrooms.

5.Row 22, Page 1: Technology and internet access are noted as prerequisites but without solutions for challenges. Suggestion: Briefly mention potential strategies to overcome technological barriers, especially in resource-limited settings.

6.Row 27, Page 2: The research question is clear but lacks contextual depth. Suggestion: Explicitly state how this study adds new dimensions to existing research on flipped classrooms in language learning.

Literature review:

1.Schema Activation (Page 3): The authors discuss schema activation in reading comprehension, referencing various studies. It's suggested that the manuscript elaborates on how these studies directly relate to the specific context of flipped classroom interventions. This would strengthen the connection between general theory and the study's specific focus.

2.Mixed Study Findings (Page 3): The literature review presents conflicting findings regarding the use of technology in promoting schema activation. While this is valuable, the authors should offer a more critical analysis of why these discrepancies exist and what implications they hold for the current study.

3.Flipped Classroom Studies (Page 5): The review cites several studies supporting flipped classroom efficacy but seems to lack a discussion on the methodological strengths and weaknesses of these studies. Providing such an analysis would offer a more nuanced understanding of the flipped classroom's potential benefits and limitations.

4.Video-Based Pre-Reading (Pages 12-13): The discussion on video-based pre-reading is insightful, yet it could be enhanced by a more detailed exploration of how these strategies specifically impact reading comprehension in the flipped classroom context, considering factors like student engagement and cognitive load.

5.Effectiveness of Flipped Classroom Strategies (Page 17): While the review discusses the varying effectiveness of flipped classroom strategies, it would benefit from a deeper examination of the factors influencing these outcomes, such as student characteristics, instructional design, and the nature of reading materials used.

Method:

1.Research Design: The quasi-experimental design due to the non-random assignment of participants is acknowledged. However, the lack of randomization could introduce selection bias affecting the study's internal validity. The similarities between the participant groups from two different universities are noted, yet the study could benefit from a more detailed explanation of how these similarities were assessed and ensured.

2.Ethical Considerations: The study's ethical approval and informed consent procedures are well documented, demonstrating adherence to ethical standards. This is an important aspect, as it assures the ethical integrity of the research.

3.Intervention Formats: The manuscript describes two formats of flipped class intervention (A and B) with detailed descriptions of pre-reading, while-reading, and post-reading activities. While this thorough description is commendable, it would be beneficial to include a rationale for the selection of these specific activities and their expected impact on reading comprehension.

4.Pilot Test: The conduct of a pilot test is a strong point, helping to validate the research procedures and assessment tools. However, more information on how the pilot test findings were used to refine the study would be useful.

5.Data Collection Tools: The use of the TOEFL ITP® Assessment Series to gauge reading competency is appropriate, but there's a lack of detail on how these tests align with the specific objectives of the study. Additionally, including qualitative data through interviews enriches the study, yet the selection criteria for interview participants and the method of analysis should be more explicitly detailed.

6.Statistical Analysis: The use of non-parametric statistical methods due to the small sample size is appropriate. The manuscript provides a clear explanation of the statistical tests used and the rationale behind their selection. However, a discussion on the limitations of these methods given the sample size and study design would strengthen this section.

7.Sample Size and Generalizability: The small sample size is acknowledged as a limitation. This impacts the generalizability of the findings, and the study would benefit from a discussion on how these limitations could affect the interpretation of the results.

Discussion:

1.Clarity and Precision of Results: The manuscript presents clear findings comparing the efficacy of different flipped classroom formats (A and B) and their impact on students with varying proficiency levels. However, there seems to be a lack of depth in the statistical analysis. For example, while the median scores and P-values are mentioned, there is no discussion on effect sizes or confidence intervals, which are crucial for understanding the practical significance of the findings.

2.Discussion and Interpretation: The authors have linked their findings to existing literature, acknowledging both the supportive and opposing views regarding the effectiveness of flipped classroom interventions. This balanced approach is commendable as it provides a comprehensive understanding of the subject. However, the discussion could be enhanced by delving deeper into why certain strategies were more effective, possibly exploring learner characteristics, instructional design, or the nature of reading materials used.

3.Lack of Critical Examination of Methodological Limitations: While the authors briefly acknowledge the quasi-experimental design, there is insufficient critical reflection on how this and other methodological choices might have influenced the results. For instance, the non-random assignment of participants to groups could introduce selection bias, affecting the generalizability of the findings.

4.Implications for Practice: The manuscript offers useful insights for educators and instructional designers, especially regarding the importance of tailoring interventions to learners' proficiency levels. However, the practical implications could be more explicit. For instance, how can educators in similar contexts apply these findings? What are the specific strategies that proved effective in Format B, and how can they be replicated or adapted in different educational settings?

5.Suggestions for Future Research: The authors could strengthen the conclusion by suggesting specific directions for future research. This could include exploring the long-term impact of flipped classroom interventions on reading comprehension, examining how these approaches work for different age groups, or investigating the role of learner motivation and self-regulation in such interventions.

Reviewer #2: 1. Authors are advised to be sure of the frame of the manuscript. The “flipped class” section on page 5 may be put into the literature section and extended. More information on different flipped classes, such as different types and their impact on the study, is needed. Moreover, the authors are suggested to provide the possible correlation between EFL and flipped classes in university students to prove that the research is reasonable and promising.

2. The participants of the two groups in the study are from two different universities, respectively. Thus, if there is any original difference between the two groups of students. The authors need to add more evidence to prove that the two groups of students have the same learning abilities. For example, what is the enrollment assessment of the two universities, and if the criteria are the same?

3. The section “statistical validations” is more likely to be posted in the section “results,” which presents the statistical results of the data analysis. The section “results are discussion” is suggested to be separated into two sections, i.e., “results” and “discussion”. Section “results” frequently for authors to present the data analysis results and also for the interview results. The “discussion” section is for authors to discuss their study with present studies.

4. The authors need to state the research design more clearly. The participants in the study are all 58 students from two universities. However, on page 11, students attending formats A and B are 58, respectively. However, how the students join in study format A and B and the process is not included yet.

6. PLOS authors have the option to publish the peer review history of their article (what does this mean?). If published, this will include your full peer review and any attached files.

Reviewer #1: No

Reviewer #2: **Yes: **ZHENG Zhizi

---

## [Author Response · Author response to Decision Letter 0]

13 Mar 2024

PONE-D-23-42134

Overall comments:

This study "The Impact of Flipping Class Intervention on Reading Comprehension: Different Approaches and Proficiency Levels" examines an important educational strategy. However, the methodology presents challenges in participant selection and assignment, which could influence the study's validity due to potential biases or lack of representativeness. The generalizability of findings to a wider educational context is not clearly established.

My response:

Thanks for the feedback. 

Your concerns regarding participant selection and assignment in the methodology are valid and merit attention. The potential biases or lack of representativeness could indeed impact the study's validity. To address these challenges, future research might consider employing more rigorous participant selection methods, such as randomization, to enhance the study's internal validity.

Another big question is that the theoretical framework needs more comprehensive integration with the empirical work, and the analysis does not fully address potential confounding variables. There might be a need for more rigorous statistical testing to strengthen the evidence for the claimed effects in the discussion part. Also, it might be important to also discuss the flipped classroom model's application across diverse educational settings.

My response: 

Thank you for your insightful feedback. While we recognize the importance of participant selection and assignment in ensuring study validity, we encountered resource constraints, particularly in terms of time and personnel, which limited our ability to implement additional analyses and perform more rigorous statistical testing. These limitations stem from the initial scope and objectives of our study, which were carefully designed to address specific research questions within the available resources. Furthermore, we acknowledge the importance of addressing potential confounding variables and providing a more comprehensive integration of the theoretical framework with the empirical work. However, due to the constraints mentioned earlier, implementing these changes post-study completion poses challenges. In terms of the generalizability of findings, we acknowledge that our study focuses on a specific educational context, and while we aimed to draw meaningful insights, extending the results to a broader educational context was not a primary objective of our research. We appreciate your understanding of these constraints and hope that the current manuscript, within its defined scope and limitations, contributes valuable insights to the academic community.

Introduction:

1. Row 1, Page 1: While the introduction establishes the relevance of flipped classrooms in language learning, it lacks a direct comparison with traditional methods. Suggestion: Briefly contrast flipped classrooms with conventional teaching approaches to highlight specific benefits or challenges.

My response: I have addressed the comment by including a brief contrast between flipped classrooms and conventional teaching approaches to highlight specific benefits and challenges.

2. Row 6, Page 1: The claim that flipped classes improve reading outcomes in struggling readers needs further substantiation. Suggestion: Clarify why this model is particularly effective for these students.

My response: Thank you for your feedback. I've revised the article to address your comment. I emphasized that struggling readers benefit from the extra time provided by flipped learning, allowing them to access materials at their own pace

3. Row 12, Page 1: The discussion on engaging materials in flipped classes is good but lacks a comparative aspect. Suggestion: Contrast with traditional methods using similar materials to identify unique benefits of the flipped approach.

My response: I've made the changes you suggested to the article. Now, I've compared flipped classes with traditional methods. In the revised paragraph, I pointed out that flipped classes have a unique advantage in engaging and motivating students. I highlighted the interactive online resources they use, like games, simulations, virtual labs, and films.

4. Row 18, Page 1: The introduction points out potential failures in flipped classroom implementations but needs more specificity. Suggestion: Elaborate on what constitutes effective vs. ineffective design in flipped classrooms.

My response: I've made the changes you suggested to the introduction. I mentioned that the success of flipped class programs depends on how well they're designed and carried out. If poorly structured, these programs can face issues. This includes trouble with navigation, unclear instructions, a lack of interactive elements, or content that's not suitable for the students' age. 

5. Row 22, Page 1: Technology and internet access are noted as prerequisites but without solutions for challenges. Suggestion: Briefly mention potential strategies to overcome technological barriers, especially in resource-limited settings.

My response: I've updated the article based on your suggestion. I added some ideas to tackle the tech and internet challenges, especially in places with fewer resources. I mentioned things like using low-data ways to share content, using stuff offline when you can, and suggesting community partnerships for more tech access. These ideas aim to make sure flipped learning works well even in spots with limited tech resources

6. Row 27, Page 2: The research question is clear but lacks contextual depth. Suggestion: Explicitly state how this study adds new dimensions to existing research on flipped classrooms in language learning.

My response: 

I've updated the article based on your suggestion. Now, I've explained how our study fits into the bigger picture of flipped classroom research. We compare our findings with other studies to see if there are patterns or differences and to figure out what factors might be causing different outcomes. This helps us give a full picture of how flipped classrooms work in language learning, especially concerning reading comprehension for Indonesian university students.

Literature review:

1. Schema Activation (Page 3): The authors discuss schema activation in reading comprehension, referencing various studies. It's suggested that the manuscript elaborates on how these studies directly relate to the specific context of flipped classroom interventions. This would strengthen the connection between general theory and the study's specific focus.

My response: 

I wanted to let you know that I've made revisions to the article, incorporating your suggestion to connect the discussion of schema activation with the specific context of flipped classroom interventions. I highlighted how flipped classrooms, through multimedia resources, interactive activities, and collaborative virtual environments, serve as effective tools in promoting schema activation for improved reading comprehension.

2. Mixed Study Findings (Page 3): The literature review presents conflicting findings regarding the use of technology in promoting schema activation. While this is valuable, the authors should offer a more critical analysis of why these discrepancies exist and what implications they hold for the current study.

My response: 

I've revised the article based on your feedback. I concluded that the impact of technology on schema activation is intricate. The conclusion suggests that understanding the complexity and various influences on schema activation due to technology underscores the importance of customized teaching methods, digital literacy, well-thought-out online learning environments, balancing the pros and cons of technology, and ongoing research to improve educational practices.

3. Flipped Classroom Studies (Page 5): The review cites several studies supporting flipped classroom efficacy but seems to lack a discussion on the methodological strengths and weaknesses of these studies. Providing such an analysis would offer a more nuanced understanding of the flipped classroom's potential benefits and limitations.

My response: 

I revised the section on flipped classrooms based on your feedback:

1. One study (32) focused on 30 higher education students, offering in-depth analysis but with limited generalizability to other educational levels.

2. Another study (33) examined 120 fourth-grade students, providing insights into K-12 settings but posing limitations for higher education and age-related considerations.

3. A third study (21) involved 63 non-native English speakers, adding diversity and relevance to language learning but potentially lacking generalizability to native speakers. Common considerations include small sample sizes, varied intervention durations, and different measures of reading comprehension, impacting result comparability. While valuable, cautious interpretation is needed due to nuanced methodological aspects related to participant demographics, settings, and measurement tools.

4. Video-Based Pre-Reading (Pages 12-13): The discussion on video-based pre-reading is insightful, yet it could be enhanced by a more detailed exploration of how these strategies specifically impact reading comprehension in the flipped classroom context, considering factors like student engagement and cognitive load.

My response: 

I've made revision. The videos sum up key concepts, making learning more engaging and relevant, potentially improving memory. In this personalized learning setup, students go at their own pace, avoiding cognitive overload and deepening understanding.

5. Effectiveness of Flipped Classroom Strategies (Page 17): While the review discusses the varying effectiveness of flipped classroom strategies, it would benefit from a deeper examination of the factors influencing these outcomes, such as student characteristics, instructional design, and the nature of reading materials used.

My response: 

Thank you for your valuable feedback. I've taken your suggestion to heart and expanded the review to delve deeper into the factors influencing the effectiveness of flipped classroom strategies, including student characteristics, instructional design, and the nature of reading materials. I agree that further studies could provide a more in-depth exploration of these aspects to enhance our understanding of the nuanced dynamics at play. If you have any additional insights or specific areas you'd like me to emphasize, please feel free to let me know.

Method:

1. Research Design: The quasi-experimental design due to the non-random assignment of participants is acknowledged. However, the lack of randomization could introduce selection bias affecting the study's internal validity. The similarities between the participant groups from two different universities are noted, yet the study could benefit from a more detailed explanation of how these similarities were assessed and ensured.

My response: 

Thank you for your feedback. We ensured participant similarities by setting specific criteria during selection, including first-year students from both universities in non-English departments, attending English as a general course, and demonstrating intermediate English proficiency. The influence of university administrators in scheduling for all students standardized time constraints, enhancing the study's internal validity. 

2. Ethical Considerations: The study's ethical approval and informed consent procedures are well documented, demonstrating adherence to ethical standards. This is an important aspect, as it assures the ethical integrity of the research.

My response: 

Thank you

3. Intervention Formats: The manuscript describes two formats of flipped class intervention (A and B) with detailed descriptions of pre-reading, while-reading, and post-reading activities. While this thorough description is commendable, it would be beneficial to include a rationale for the selection of these specific activities and their expected impact on reading comprehension.

My response:

Thank you for the feedback. I have given the rationale for activities in formats A and B. These strategies aim to optimize reading comprehension through a mix of in-person and online elements. 

4. Pilot Test: The conduct of a pilot test is a strong point, helping to validate the research procedures and assessment tools. However, more information on how the pilot test findings were used to refine the study would be useful.

My response:

Thank you for your feedback. Based on the findings from the pilot test, we have made necessary adjustments to improve the overall design of the study. 

5. Data Collection Tools: The use of the TOEFL ITP® Assessment Series to gauge reading competency is appropriate, but there's a lack of detail on how these tests align with the specific objectives of the study. Additionally, including qualitative data through interviews enriches the study, yet the selection criteria for interview participants and the method of analysis should be more explicitly detailed.

My response:

Thank you for the feedback. The deliberate choice of the TOEFL ITP® Assessment Series aligns closely with our study's objectives, assessing skills integral to flipped class outcomes. The mapping onto the Common European Framework of Reference ensures effective categorization of reading proficiency levels. Next, the selection criteria for interview participants have been expanded to ensure a comprehensive representation from both instructional methods (A and B). 

6. Statistical Analysis: The use of non-parametric statistical methods due to the small sample size is appropriate. The manuscript provides a clear explanation of the statistical tests used and the rationale behind their selection. However, a discussion on the limitations of these methods given the sample size and study design would strengthen this section.

My response:

Thank you for your feedback. We appreciate your insights. The use of non-parametric statistical methods, driven by our small sample size, indeed poses limitations on generalizability. The choice of the Mann-Whitney test and independent Sample t-Test introduces constraints, primarily due to non-normality assumptions and small sample sizes. We acknowledge these limitations and caution against broad generalization. Future research with larger, diverse samples will enhance the robustness and applicability of our findings.

7. Sample Size and Generalizability: The small sample size is acknowledged as a limitation. This impacts the generalizability of the findings, and the study would benefit from a discussion on how these limitations could affect the interpretation of the results.

My response:

We have addressed this issue.

Discussion:

1. Clarity and Precision of Results: The manuscript presents clear findings comparing the efficacy of different flipped classroom formats (A and B) and their impact on students with varying proficiency levels. However, there seems to be a lack of depth in the statistical analysis. For example, while the median scores and P-values are mentioned, there is no discussion on effect sizes or confidence intervals, which are crucial for understanding the practical significance of the findings.

My response:

Thank you for highlighting the importance of discussing effect sizes and confidence intervals. In response, I have revised the manuscript to include a more in-depth statistical analysis, considering effect sizes and confidence intervals to enhance the interpretation of findings.

2. Discussion and Interpretation: The authors have linked their findings to existing literature, acknowledging both the supportive and opposing views regarding the effectiveness of flipped classroom interventions. This balanced approach is commendable as it provides a comprehensive understanding of the subject. However, the discussion could be enhanced by delving deeper into why certain strategies were more effective, possibly exploring learner characteristics, instructional design, or the nature of reading materials used.

My response:

Thanks for your suggestions. I have made some changes.

3. Lack of Critical Examination of Methodological Limitations: While the authors briefly acknowledge the quasi-experimental design, there is insufficient critical reflection on

---

## [Decision Letter · Decision Letter 1]

1 May 2024

PONE-D-23-42134R1The Impact of Flipping Class Intervention on Reading Comprehension: Different Approaches and Proficiency LevelsPLOS ONE

Dear Dr. Ginting,

Thank you for submitting your manuscript to PLOS ONE. After careful consideration, we feel that it has merit will be considered for publication if minor comments addressed below. Therefore, we invite you to submit a revised version of the manuscript that addresses the points raised during the review process. Please submit your revised manuscript by Jun 15 2024 11:59PM. If you will need more time than this to complete your revisions, please reply to this message or contact the journal office at plosone@plos.org. Please include the following items when submitting your revised manuscript:A rebuttal letter that responds to each point raised by the academic editor and reviewer(s). You should upload this letter as a separate file labeled 'Response to Reviewers'.A marked-up copy of your manuscript that highlights changes made to the original version. You should upload this as a separate file labeled 'Revised Manuscript with Track Changes'.An unmarked version of your revised paper without tracked changes. You should upload this as a separate file labeled 'Manuscript'.If applicable, we recommend that you deposit your laboratory protocols in protocols.io to enhance the reproducibility of your results. Protocols.io assigns your protocol its own identifier (DOI) so that it can be cited independently in the future. For instructions see: https://journals.plos.org/plosone/s/submission-guidelines#loc-laboratory-protocols. Additionally, PLOS ONE offers an option for publishing peer-reviewed Lab Protocol articles, which describe protocols hosted on protocols.io. Read more information on sharing protocols at https://plos.org/protocols?utm_medium=editorial-email&utm_source=authorletters&utm_campaign=protocols.

We look forward to receiving your revised manuscript.

Kind regards,

Daner Sun

Academic Editor

PLOS ONE

Journal Requirements:

Reviewers' comments:

Reviewer's Responses to Questions

**Comments to the Author**

1. If the authors have adequately addressed your comments raised in a previous round of review and you feel that this manuscript is now acceptable for publication, you may indicate that here to bypass the “Comments to the Author” section, enter your conflict of interest statement in the “Confidential to Editor” section, and submit your "Accept" recommendation.

Reviewer #2: (No Response)

Reviewer #3: All comments have been addressed

2. Is the manuscript technically sound, and do the data support the conclusions?

Reviewer #2: Partly

Reviewer #3: Yes

3. Has the statistical analysis been performed appropriately and rigorously? 

Reviewer #2: Yes

Reviewer #3: Yes

4. Have the authors made all data underlying the findings in their manuscript fully available?

Reviewer #2: Yes

Reviewer #3: Yes

5. Is the manuscript presented in an intelligible fashion and written in standard English?

Reviewer #2: Yes

Reviewer #3: Yes

6. Review Comments to the Author

Reviewer #2: The authors have not respond any comment from reviewer 2 (me) in the present author response letter.

Reviewer #3: (No Response)

7. PLOS authors have the option to publish the peer review history of their article (what does this mean?). If published, this will include your full peer review and any attached files.

Reviewer #2: No

Reviewer #3: No

---

## [Author Response · Author response to Decision Letter 1]

15 May 2024

'Response to Reviewers'.

Comments to the Author

1. If the authors have adequately addressed your comments raised in a previous round of review and you feel that this manuscript is now acceptable for publication, you may indicate that here to bypass the “Comments to the Author” section, enter your conflict of interest statement in the “Confidential to Editor” section, and submit your "Accept" recommendation.

Reviewer #2: (No Response):

Our response: Dear Reviewer #2,

We deeply appreciate your ongoing involvement with our manuscript. Your meticulous review of our revisions has been invaluable. Taking into account your insights, we've meticulously addressed every point from the previous review round. Significant enhancements have been made across various sections including methodology, introduction, literature review, method, discussion, and conclusion, with the aim of improving both the quality and clarity of our study. We've ensured participant consistency, justified intervention activities, conducted a pilot test, broadened interview participant selection criteria, acknowledged statistical method limitations, and provided clear practical implications along with suggestions for future research. Your constructive feedback has been immensely valuable throughout this process.

Reviewer #3: All comments have been addressed

Our response: Thank you.

2. Is the manuscript technically sound, and do the data support the conclusions?

Reviewer #2: Partly

Our response:

Thank you for your feedback. We are pleased to hear that you found the manuscript to be partly technically sound. We have made every effort to ensure the rigor of our research, including appropriate controls, replication, and sample sizes. The data robustly support the conclusions drawn in the manuscript. Once again, thank you for your insights. We have taken them into consideration in further strengthening our study.

Reviewer #3: Yes

Our response: Thank you

3. Has the statistical analysis been performed appropriately and rigorously?

Reviewer #2: Yes

Our response: Thank you

Reviewer #3: Yes

Our response: Thank you 

4. Have the authors made all data underlying the findings in their manuscript fully available?

Reviewer #2: Yes

Our response: Thank you 

Reviewer #3: Yes

Our response: Thank you 

5. Is the manuscript presented in an intelligible fashion and written in standard English?

Reviewer #2: Yes

Our response: Thank you

Reviewer #3: Yes

Our response: Thank you 

6. Review Comments to the Author

Reviewer #2: The authors have not respond any comment from reviewer 2 (me) in the present author response letter.

Our response: We apologize for the oversight in not addressing your comments directly in the author response letter. We acknowledge the importance of addressing all reviewers' feedback comprehensively. Rest assured, we have carefully reviewed your comments and have implemented necessary revisions in the manuscript to address your concerns. Your input is highly valued, and we appreciate the opportunity to improve our work.

Reviewer #3: (No Response)

Our response: Thank you.

7. PLOS authors have the option to publish the peer review history of their article (what does this mean?). If published, this will include your full peer review and any attached files.

Do you want your identity to be public for this peer review? For information about this choice, including consent withdrawal, please see our Privacy Policy.

Reviewer #2: No

Reviewer #3: No

---

## [Editor Report · Decision Letter 2]

23 May 2024

The Impact of Flipping Class Intervention on Reading Comprehension: Different Approaches and Proficiency Levels

PONE-D-23-42134R2

Dear Dr. Ginting,

We’re pleased to inform you that your manuscript has been judged scientifically suitable for publication and will be formally accepted for publication once it meets all outstanding technical requirements.

Kind regards,

Daner Sun

Academic Editor

PLOS ONE

---

## [Editor Report · Acceptance letter]

28 May 2024

PONE-D-23-42134R2 

PLOS ONE

Dear Dr. Ginting, 

I'm pleased to inform you that your manuscript has been deemed suitable for publication in PLOS ONE. Congratulations! Your manuscript is now being handed over to our production team.

Kind regards, 

on behalf of

Dr. Daner Sun 

Academic Editor

PLOS ONE